# Exploring the Potential of Sulfur Moieties in Compounds Inhibiting Steroidogenesis

**DOI:** 10.3390/biom13091349

**Published:** 2023-09-05

**Authors:** Tomasz M. Wróbel, Katyayani Sharma, Iole Mannella, Simonetta Oliaro-Bosso, Patrycja Nieckarz, Therina Du Toit, Clarissa Daniela Voegel, Maria Natalia Rojas Velazquez, Jibira Yakubu, Anna Matveeva, Søren Therkelsen, Flemming Steen Jørgensen, Amit V. Pandey, Agnese C. Pippione, Marco L. Lolli, Donatella Boschi, Fredrik Björkling

**Affiliations:** 1Department of Synthesis and Chemical Technology of Pharmaceutical Substances, Medical University of Lublin, Chodźki 4a, 20093 Lublin, Poland; 2Department of Drug Design and Pharmacology, University of Copenhagen, Jagtvej 160, 2100 Copenhagen, Denmark; 3Department of Pediatrics, Division of Endocrinology, Diabetology and Metabolism, University Children’s Hospital, University of Bern, 3010 Bern, Switzerland; 4Translational Hormone Research Program, Department of Biomedical Research, University of Bern, 3010 Bern, Switzerland; 5Graduate School for Cellular and Biomedical Sciences, University of Bern, 3012 Bern, Switzerland; 6Department of Drug Science and Technology, University of Turin, 10125 Turin, Italy; 7Department of Nephrology and Hypertension, University Hospital Inselspital, University of Bern, 3010 Bern, Switzerland

**Keywords:** CYP17A1, AKR1C3, prostate cancer, enzyme inhibition

## Abstract

This study reports on the synthesis and evaluation of novel compounds replacing the nitrogen-containing heterocyclic ring on the chemical backbone structure of cytochrome P450 17α-hydroxylase/12,20-lyase (CYP17A1) inhibitors with a phenyl bearing a sulfur-based substituent. Initial screening revealed compounds with marked inhibition of CYP17A1 activity. The selectivity of compounds was thereafter determined against cytochrome P450 21-hydroxylase, cytochrome P450 3A4, and cytochrome P450 oxidoreductase. Additionally, the compounds showed weak inhibitory activity against aldo-keto reductase 1C3 (AKR1C3). The compounds’ impact on steroid hormone levels was also assessed, with some notable modulatory effects observed. This work paves the way for developing more potent dual inhibitors specifically targeting CYP17A1 and AKR1C3.

## 1. Introduction

Cancer remains a major health problem and is the second leading cause of death. Of particular concern is the rising incidence of prostate cancer (PCa) from 2014 to 2019 after two decades of decline [1]. Despite significant advancements in treatment options for PCa, there is still an urgent need for the development of more effective therapies. Advanced disease can progress into castration-resistant prostate cancer (CRPC) which is characterized by poor prognosis [2]. The androgen receptor (AR) signaling pathway plays a crucial role in the development and progression of PCa [3]. Androgens, such as testosterone (T) and dihydrotestosterone (DHT), bind to the AR and activate downstream signaling pathways, leading to the growth and survival of PCa cells. Therefore, targeting the AR signaling pathway has been the focus of therapeutic strategies for PCa. Among the potential targets in prostate cancer, cytochrome P450 17α-hydroxylase/17,20-lyase (CYP17A1) is a key enzyme involved in the biosynthesis of androgens. It catalyzes two essential reactions in androgen biosynthesis: a 17α-hydroxylation and a 17,20-lyase reaction (Figure 1). The inhibition of CYP17A1 results in decreased androgen production and has emerged as a promising strategy for the treatment of PCa (Figure 2) [4]. CYP17A1 inhibitors can be classified into two categories: steroidal represented by abiraterone, and nonsteroidal [5]. Abiraterone is the only CYP17A1 inhibitor that has been approved as a drug for clinical use in the treatment of CRPC. However, the development of resistance to abiraterone occurs with time, and, therefore, alternative CYP17A1 inhibitors might circumvent this issue. Nonsteroidal inhibitors, such as seviteronel or orteronel, have shown promising results in preclinical studies [6,7]. Unfortunately, none of these compounds have reached clinical practice [8].

PCa progression and aggressiveness have also been associated with elevated levels of the aldo-keto reductase 1C3 (AKR1C3) enzyme, another important enzyme in the steroidogenesic pathway to potent androgens (Figure 1) [9,10]. Recent studies have shown that PCa cells grown in the presence of abiraterone acquire resistance to this drug through increasing intracrine androgen synthesis and overexpressing *AKR1C3*. This discovery provided a preclinical proof-of-principle for investigating the combination of targeting AKR1C3 drugs (e.g., indomethacin) with abiraterone or other cancer chemotherapeutics for CRPC treatment [11,12].

The common structural feature of nearly all CYP17A1 inhibitors is the presence of a sp^2^-hybridized nitrogen atom in a heteroaromatic system. The lone pair on the nitrogen coordinates with the iron in the heme prosthetic group, thus blocking enzyme activity. Pyridine and imidazole moieties have been used extensively for that purpose because they offer the most favorable binding energy based on density functional theory (DFT) calculations [13,14]. Other nitrogen-bearing heterocycles have found their way into the design of many CYP17A1 inhibitors. However, they are less common. While the field appears to be biased towards the utilization of the nitrogen, other atoms can be potentially used for the purpose of coordination with the heme iron. For instance, curcumin and theaflavin have been described as capable of CYP17A1 inhibition [15,16]. Both compounds do not contain nitrogen atoms and can only coordinate to the heme via oxygen atoms. When curcumin was docked to the PDB ID: 3RUK model, it exhibited a similar binding pose to steroid substrates, with its phenolic oxygen positioned 2.4 Å away from the heme iron [15,16]. There are also examples of the sulfur atom interacting with the heme. The bacterial CYP BM3 (CYP102A1) mutant M11 structure displays the anionic form of the sulfhydryl group in dithiothreitol coordinating to the heme with a 2.3 Å Fe-S distance [17]. A few nitric oxide synthase inhibitors based on thioethers have demonstrated type II binding to the heme group, with some sulfur atoms in these inhibitors coordinating to the heme iron with Fe-S distances of approximately 2.7 Å, as observed in X-ray crystallography [18,19].

Except for limited efforts of introducing sulfur-containing moieties into a steroidal CYP17A1 inhibitor, to the best of our knowledge, there have been no similar attempts to design non-steroidal inhibitors [20]. Therefore, we were interested in probing potential interactions between sulfenyl and sulfinyl groups with the heme iron of CYP17A1. In the current work, we present the design, synthesis, and evaluation of sulfur-based CYP17A1 non-steroidal inhibitors.

## 2. Materials and Methods

### 2.1. Chemistry

All reagents were obtained from commercial sources and used as received without additional purification. NMR spectra were acquired using a Bruker AVANCE III 600 MHz spectrometer equipped with a BBO broadband probe and referenced to either TMS or residual solvent peak. HRMS spectra were recorded using a Bruker microTOF-Q II spectrometer. HPLC analysis was run on a Thermo-Fisher Ultimate 3000 RS chromatograph using a generic gradient of 20% to 100% acetonitrile in water on a C18 column.

### 2.2. General Procedure to Obtain Compounds ***3a*** to ***3d***

A dry vial was evacuated/backfilled with argon three times and charged with 3-bromothioanisole **2a** or 4-bromothioanisole **2b** (223 mg, 1.10 mmol), t-BuXPhos Pd G3 (40 mg, 0.05 mmol), t-BuXPhos (21 mg, 0.05 mmol), benzimidazole **1a** or indole **1b** (246 mg, 1 mmol), and t-BuONa (240 mg, 2.5 mmol). For indole derivatives, the reaction was conducted in t-BuOH (3 mL) and in the case of benzimidazole derivatives, THF (2 mL) was used. The reaction was microwaved at 100 °C for 1 h. It was then diluted with EtOAc (25 mL) and filtered through a silica plug. The crude product was purified via flash chromatography on silica as indicated below.

#### 2.2.1. *N*-(4-(1H-Benzo[d]imidazol-1-yl)phenyl)-3-(methylthio)aniline (**3a**)

Elution with EtOAc/heptane yielded orange oil (81 mg, 24%) which can be triturated with EtOAc and recrystallized from MeOH to obtain solid material. ^1^H NMR (600 MHz, DMSO) δ 8.54 (s, 1H), 8.46 (s, 1H), 7.79–7.74 (m, 1H), 7.56 (dt, *J* = 8.1, 0.9 Hz, 1H), 7.54–7.50 (m, 2H), 7.34–7.25 (m, 4H), 7.22 (t, *J* = 7.9 Hz, 1H), 7.00 (t, *J* = 2.0 Hz, 1H), 6.95 (ddd, *J* = 8.1, 2.3, 0.9 Hz, 1H), 6.78 (ddd, *J* = 7.8, 1.8, 0.9 Hz, 1H), 2.47 (s, 3H). ^13^C NMR (151 MHz, DMSO) δ 143.5, 143.3, 143.2, 142.9, 139.0, 133.5, 129.7, 127.7, 125.1, 123.1, 122.1, 119.7, 117.8, 117.4, 114.3, 113.6, 110.5, 14.6. HRMS (ESI) [M+H]^+^ calc. 332.1216, exp. 332.1216.

#### 2.2.2. *N*-(4-(1H-Benzo[d]imidazol-1-yl)phenyl)-4-(methylthio)aniline (**3b**)

Elution with EtOAc/heptane and trituration with ether/hexane yielded a beige solid (19 mg, 6%) which can be further recrystallized from EtOAc/hexane. ^1^H NMR (600 MHz, DMSO) δ 8.50 (s, 1H), 8.45 (s, 1H), 7.78–7.74 (m, 1H), 7.56–7.53 (m, 1H), 7.51–7.47 (m, 2H), 7.33–7.28 (m, 2H), 7.28–7.21 (m, 5H), 7.16–7.11 (m, 2H), 2.44 (s, 3H). ^13^C NMR (151 MHz, DMSO) δ 143.5, 143.3, 140.6, 133.6, 128.9, 128.0, 127.3, 125.1, 123.1, 122.0, 119.7, 118.3, 116.8, 110.5, 16.5 HRMS (ESI) [M+H]^+^ calc. 332.1216, exp. 332.1220.

#### 2.2.3. *N*-(4-(1H-Indol-1-yl)phenyl)-3-(methylthio)aniline (**3c**)

Elution with EtOAc/heptane yielded red oil (72 mg, 22%) which can be recrystallized from MeOH by slow evaporation to obtain solid material. ^1^H NMR (600 MHz, CDCl_3_) δ 7.67 (ddd, *J* = 7.8, 1.3, 0.7 Hz, 1H), 7.49 (dt, *J* = 8.2, 0.9 Hz, 1H), 7.34–7.29 (m, 2H), 7.25 (d, *J* = 3.2 Hz, 1H), 7.22–7.12 (m, 3H), 7.12–7.07 (m, 2H), 6.95 (t, *J* = 2.0 Hz, 1H), 6.82 (dt, *J* = 7.8, 2.0 Hz, 2H), 6.64 (dt, *J* = 3.2, 0.7 Hz, 1H), 5.73 (s, 1H), 2.43 (d, *J* = 0.6 Hz, 3H). ^13^C NMR (151 MHz, CDCl_3_) δ 143.4, 141.5, 139.9, 136.3, 133.2, 129.9, 129.1, 128.3, 125.8, 122.3, 121.2, 120.3, 119.3, 118.9, 115.7, 114.8, 110.6, 103.1, 15.8. HRMS (ESI) [M+H]^+^ calc. 331.1263, exp. 331.1269.

#### 2.2.4. *N*-(4-(1H-Indol-1-yl)phenyl)-4-(methylthio)aniline (**3d**)

Elution with EtOAc/heptane yielded clear oil (13 mg, 4%) which can be recrystallized from MeOH to obtain solid material. ^1^H NMR (600 MHz, CDCl_3_) δ 7.68 (dt, *J* = 7.8, 1.0 Hz, 1H), 7.49 (dd, *J* = 8.3, 1.0 Hz, 1H), 7.37–7.33 (m, 2H), 7.30–7.24 (m, 3H), 7.21 (ddd, *J* = 8.3, 7.0, 1.3 Hz, 1H), 7.17–7.11 (m, 3H), 7.07 (d, *J* = 8.1 Hz, 2H), 6.65 (dd, *J* = 3.2, 0.8 Hz, 1H), 5.96 (s, 0H), 2.47 (s, 3H). ^13^C NMR (151 MHz, CDCl_3_) δ 141.7, 140.6, 136.2, 133.0, 130.1, 129.6, 129.0, 128.2, 125.8, 122.1, 121.0, 120.1, 119.1, 118.2, 110.4, 102.9, 17.6. HRMS (ESI) [M+H]^+^ calc. 331.1263, exp. 331.1270.

### 2.3. General Procedure to Obtain Compounds ***4a*** to ***4d***

To a stirred solution of the sulfide (1 eq) in phenol (12 eq) at 40 °C, 30% aq. H_2_O_2_ (2 eq) was added and the reaction was stirred for 5 min. The excess H_2_O_2_ was quenched with saturated aq. Na_2_SO_3_ solution, and the phenol was neutralized with 10% aq. NaOH. The product was extracted three times with EtOAc, and the combined organics were washed with aq. NaOH and dried with MgSO_4_. The crude product was purified using flash chromatography on silica, as described below.

#### 2.3.1. *N*-(4-(1H-Benzo[d]imidazol-1-yl)phenyl)-3-(methylsulfinyl)aniline (**4a**)

Elution with EtOAc yielded a white solid (44mg, 39%). ^1^H NMR (600 MHz, DMSO) δ 8.85 (s, 1H), 8.49 (s, 1H), 7.79–7.75 (m, 1H), 7.59–7.54 (m, 3H), 7.49–7.42 (m, 2H), 7.36–7.25 (m, 5H), 7.11 (ddd, *J* = 7.7, 1.7, 0.9 Hz, 1H), 2.76 (s, 3H). ^13^C NMR (151 MHz, DMSO) δ 148.1, 144.5, 144.1, 143.8, 142.7, 134.0, 130.7, 128.9, 125.7, 123.7, 122.7, 120.3, 118.8, 118.6, 115.1, 111.1, 111.0, 43.8. HRMS (ESI) [M+H]^+^ calc. 348.1165, exp. 348.1177.

#### 2.3.2. *N*-(4-(1H-Benzo[d]imidazol-1-yl)phenyl)-4-(methylsulfinyl)aniline (**4b**)

Elution with MeOH/DCM yielded a white solid (30 mg, 15%). ^1^H NMR (600 MHz, DMSO) δ 8.93 (s, 1H), 8.49 (s, 1H), 7.81–7.75 (m, 1H), 7.62–7.54 (m, 5H), 7.39–7.35 (m, 2H), 7.34–7.26 (m, 4H), 2.71 (s, 3H). ^13^C NMR (151 MHz, DMSO) δ 146.2, 144.1, 143.8, 142.3, 136.0, 134.0, 129.2, 126.0, 125.6, 123.8, 122.7, 120.3, 119.2, 116.7, 111.1, 43.6. HRMS (ESI) [M+H]^+^ calc. 348.1165, exp. 348.1178.

#### 2.3.3. *N*-(4-(1H-Indol-1-yl)phenyl)-3-(methylsulfinyl)aniline (**4c**)

Elution with MeOH/DCM and trituration with ether yielded a solid product (37 mg, 16%). ^1^H NMR (600 MHz, DMSO) δ 8.78 (s, 1H), 7.65 (d, *J* = 7.8 Hz, 1H), 7.57 (d, *J* = 3.2 Hz, 1H), 7.54–7.39 (m, 5H), 7.36–7.30 (m, 2H), 7.29–7.23 (m, 1H), 7.23–7.15 (m, 1H), 7.14–7.05 (m, 2H), 6.67 (d, *J* = 3.3 Hz, 1H), 2.75 (s, 3H). ^13^C NMR (151 MHz, DMSO) δ 148.0, 144.9, 141.5, 135.9, 132.4, 130.7, 129.2, 129.1, 125.7, 122.5, 121.3, 120.4, 118.9, 118.4, 114.7, 110.8, 110.6, 103.3, 43.8. HRMS (ESI) [M+H]^+^ calc. 347.1213, exp. 347.1208

#### 2.3.4. *N*-(4-(1H-Indol-1-yl)phenyl)-4-(methylsulfinyl)aniline (**4d**)

Elution with EtOAc/heptane and trituration with ether yielded a pale yellow solid (30 mg, 32%). ^1^H NMR (600 MHz, DMSO) δ 8.81 (s, 1H), 7.68–7.63 (m, 1H), 7.61–7.56 (m, 3H), 7.53–7.47 (m, 3H), 7.36–7.32 (m, 2H), 7.29–7.24 (m, 2H), 7.19 (ddd, *J* = 8.3, 7.0, 1.3 Hz, 1H), 7.12 (ddd, *J* = 7.9, 7.0, 1.0 Hz, 1H), 6.67 (dd, *J* = 3.3, 0.8 Hz, 1H), 2.71 (s, 3H). ^13^C NMR (151 MHz, DMSO) δ 146.6, 141.1, 135.8, 135.6, 132.8, 129.2, 129.1, 126.0, 125.6, 122.5, 121.3, 120.4, 119.6, 116.3, 110.8, 103.4, 43.6. HRMS (ESI) [M+H]^+^ calc. 347.1213, exp. 347.1227

### 2.4. Biology

Chemicals: Trilostane was obtained from the extraction of the commercially available tablets Modrenal^®^ (Bioenvision, New York, NY, USA). Abiraterone acetate was purchased from MedChemExpress^®^, Lucerna Chem AG (Lucerne, Switzerland). Radiolabeled substrates, Progesterone [4-^14^C] (SA 55mCi/mmol; Conc. 0.1mCi/mL); 17α-Hydroxypregnenolone [21-^3^H] (SA 15Ci/mmol; Conc. 1 mCi/mL) and 17α-Hydroxyprogesterone [1, 2, 6, 7-^3^H] (SA 60 Ci/mmol; Conc. 1 mCi/mL) were obtained from American Radiolabeled Chemicals Inc. (St. Louis, MO, USA). Non-radiolabeled standard substrates, Progesterone, 17α-Hydroxypregnenolone, and 17α-Hydroxyprogesterone; 3-(4,5-Dimethyl-2-thiazolyl)-2,5-diphenyl-2H-tetrazolium bromide (MTT); Dimethyl sulfoxide (DMSO) and Dextran were purchased from Sigma-Aldrich^®^ (St. Louis, MO, USA). Organic solvents such as Isooctane, Ethyl acetate, and Chloroform/Trichloromethane were acquired from Carl Roth^®^ GmbH + Co. KG (Karlsruhe, Germany). Activated Charcoal was obtained from Merck AG (Darmstadt, Germany). For the mass spectrometric analysis of steroids, all LC-MS grade solvents were from Biosolve (Valkensaard, The Netherlands) and steroid standards were from obtained as certified reference solutions from Cerillant (Round Rock, TX, USA) or from Steraloids (Newport, RI, USA).

Cell lines and culture: The human adrenocortical carcinoma cell line NCI H295R was obtained from the American Type Culture Collection (ATCC^®^ CRL2128™), Manassas, VA, USA [21]. Cells between passages 12 and 24 were cultivated in DMEM/Ham’s F-12 medium (1:1 Mix) supplemented with L-glutamine and 15 mM HEPES (Gibco™, Thermo Fisher Scientific, Waltham, MA, USA) along with 5% Nu-Serum I; 0.1% insulin, transferrin, selenium in the form of ITS Premix (Corning™, Manassas, VA, USA) and 1% penicillin–streptomycin (Gibco™, Thermo Fisher Scientific, Waltham, MA, USA) at 37 °C in a humid atmosphere with a constant supply of 5% carbon dioxide to maintain the physiological Ph [22]. The Lymph Node Carcinoma of the Prostate (LNCaP) clone FGC cell line was purchased from American Type Culture Collection (ATCC) (ATCC^®^: CRL-1740™). The cell line was grown and maintained in vitro with Roswell Park Memorial Institute (RPMI) 1640 medium (Gibco™, Thermo Fisher Scientific, Waltham, MA, USA) supplemented with 10% fetal bovine serum (FBS) and 1% penicillin (100 U/mL)–streptomycin (100 µg/mL).

Cell Viability assays: To determine the effect of test compounds on the cellular activity of LNCaP and NCI H295R cells, a 3-(4,5-Dimethylthiazol-2-yl)-2,5-Diphenyltetrazolium Bromide (MTT)-based cell viability assay was performed. For the NCI-H295R cell assays, in a 96-well plate, about 30,000 cells per well were seeded with complete medium. The next day, the medium was replaced with fresh medium and 10 µM of test compounds were added. DMSO (less than 1% *v*/*v*) was used as a vehicle control. Abiraterone was used as a positive control. In total, 0.5mg/mL MTT reagent was added to the culture medium for another 4 h. After the incubation, the medium was entirely replaced with DMSO to dissolve the formazan crystals. After 20 min, absorbance was measured at 570nm (SpectraMax M2, Bucher Biotec, Basel, Switzerland). Percent viability was calculated with respect to the mean value of control samples.

The prostate cancer LNCAP cells were seeded in 96-well culture plates at a density of 0.5 × 10^4^ cells per well and grown overnight at normoxic conditions with 5% CO_2,_ temperature at 37 °C, and humidity at 90%. After 24 h, the medium was changed, and dilutions of the drugs were added to the medium and the cells were incubated for another 24h or 48 h. After the 24 h or 48 h incubation of cells with the drugs, 20 µL of sterile filtered 5 mg/mL MTT in PBS solvent was added into each well, and the incubation was continued for another 2 h. After the incubation with MTT, the culture medium in each well was replaced with 200 µL of DMSO, and the plate was incubated for 20 min in the dark. The absorbances of the solution in each well were then measured at 570 nm. The percentage cell viability was calculated as previously described [15].

CYP17A1 and CYP21A2 Enzyme activity assay: According to our previously established protocols [15], NCI H295R cells were seeded overnight in a 12-well plate at a cell density of 0.5 × 10^6^ cells per well. Overall, 10 µM of test compounds were added to respective wells containing fresh medium and were incubated for 4 h. Abiraterone and DMSO were used as the reference and control, respectively. To determine CYP17A1 hydroxylase activity, cells were treated with the substrate, [^14^C]-Progesterone, at a concentration of 10,000 cpm/1 µM per well [23]. Trilostane was added prior to the addition of test compounds in order to block 3β-hydroxysteroid dehydrogenase activity [24]. Radiolabeled steroids were extracted from the media with ethyl acetate and isooctane (1:1 *v*/*v*) and separated through Thin Layer Chromatography (TLC) on a silica-gel-coated aluminum plate (Supelco^®^ Analytics, Sigma Aldrich Chemie GmbH, Darmstadt, Germany). TLC spots were exposed to a phosphor screen and detected via autoradiography using Typhoon™ FLA-7000 PhosphorImager (GE Healthcare, Uppsala, Sweden). Radioactivity was quantified using ImageQuant™ TL analysis software (GE Healthcare Europe GmbH, Freiburg, Germany). Enzyme activity was calculated as a percentage of radioactivity incorporated into the product with respect to the total radioactivity. Similarly, CYP21A2 activity was evaluated using [^3^H]-17α-Hydroxyprogesterone (~30,000 cpm/1 µM per well) as the substrate [25].

Using similar treatment conditions, [21-^3^H]-17α-Hydroxypregnenolone (50,000 cpm/1 µM per well) was used as a substrate to analyze CYP17A1 lyase activity. Tritiated water release assays were performed to measure DHEA production [26]. Steroids in the media were precipitated using 5% activated charcoal/0.5% dextran solution. The enzyme activity was estimated with reference to the water-soluble tritiated by-product formed in an equimolar ratio with the corresponding steroid product. The radioactivity in the aqueous phase was measured via Liquid Scintillation counting (MicroBeta2^®^ Plate Counter, PerkinElmer Inc. Waltham, MA, USA). The percent inhibition was calculated with respect to the control [27].

Steroid profiling using mass spectrometry: Steroid profiling in NCI H295R cells was performed using a liquid chromatography high-resolution mass spectrometry (LC-HRMS) method as previously described and validated with 1 µM of the unlabeled substrate added to the cell culture [28,29,30]. Steroids from 500 µL cell media, plus 38 µL of a mixture of internal standards (at 3.8 nM each), were extracted using solid-phase extraction with an OasisPrime HLB 96-well plate. Samples were resuspended in 100 µL 33% methanol in water and 20 µL injected into the LC-HRMS instrument (Vanquish UHPLC coupled to a QExactive Orbitrap Plus, from Thermo Fisher Scientific) using an Acquity UPLC HSS T3 column (Waters, Milford, MA, USA). Data from the mass spectrometer were processed using TraceFinder 4.0 (Thermo Fisher Scientific, Waltham, MA, USA).

AKR1C3 inhibitor screening: AKR1C3 expression and purification were performed as previously described [31]. Briefly, the bacteria cells were grown in YT2X media that was supplemented with ampicillin, and, at OD 600 nm = 0.6, the expression was induced by IPTG 0.5 mM for 2 h. The bacteria were then centrifuged and lysed with four freeze–thaw cycles in the presence of lysozyme and protease inhibitors. The lysate was centrifuged for 30 min at 13,000× *g* and the supernatant was collected. AKR1C3 was affinity purified via an N-terminal GST-tag on glutathione (GT) sepharose (GE-Healthcare) and cleaved by thrombin, according to the manufacturer’s protocol.

The inhibition assays were performed on purified recombinant enzyme, as previously described [31]. Briefly, the enzymatic reaction was fluorometrically (exc/em; 340 nm/460 nm) monitored by the measurement of NADPH production on the Ensight plate reader (Perkin Elmer, Waltham, MA, USA) at 37 °C. The assay mixture, which contained 100 mM phosphate buffer pH 7, 200 μM NADP^+^, S-tetralol (in ETOH), inhibitor (in DMSO), and purified enzyme (1 μM), was added to a 96-well plate at a final volume of 200 μL. The S-tetralol concentration was 160 μM, in accordance with the Km described for AKR1C3 under the same experimental conditions. The solvent added to the reaction mixture did not exceed 10% of the final concentration. Percentage inhibition with respect to the controls that contained the same amount of solvent, without the inhibitor, was calculated from the initial velocities, which were obtained via the linear regression of the progress curve at different inhibitor concentrations. The results are expressed as the mean value ± standard error (SE) of at least three experiments, each carried out in triplicate.

Statistical analysis: Calculations were performed using Microsoft Excel and GraphPad Prism 3.0 (Graph Pad Software, Inc. San Diego, CA, USA). Data are represented as the mean of triplicate values. Dunnett’s multiple comparison ANOVA test was performed to determine the significant difference between the mean values of samples and the control. Error bars denote standard deviation from respective mean values. Significant *p* values were set as * *p* < 0.05 and ** *p* < 0.01, *** *p* < 0.001.

### 2.5. Computational Chemistry

The experimentally determined structure of CYP17A1 complexed with galeterone (PDB 3SWZ) was used as a target for the docking studies due to the structural similarity between galeterone and the compounds reported in this paper [32]. The protein was prepared using the Protein Preparation Wizard in Maestro (v. 11.1) [33]. The Protein Preparation Wizard optimizes the hydrogen-bonding network at pH 7.0 and performs a short restrained protein minimization using the OPLS-2005 force field [34]. The ligands were prepared by the LigPrep module in Maestro [33]. Docking was performed with GOLD (Genetic Optimisation for Ligand Docking) version 5.6 [35]. The binding site was defined to be within 15 Å around the Fe atom in the heme group. Ligands were docked 10 times with the default option slow genetic algorithm, and with the heme-modified ChemScore as scoring function [36]. The sulfoxides may exist in two enantiomeric forms with an R- or S-configuration at the sulfur atom, and, accordingly, both stereoisomers were docked to CYP17A1.

## 3. Results and Discussion

### 3.1. Chemistry

In our previous work, we synthesized non-steroidal ligands of CYP17A1 which could only bind to the heme via the nitrogen atom [37]. Here, we were interested in using sulfur-based moieties in order to discern if they could constitute a viable binding group. Thus, we envisioned using sulfides and sulfoxides for this purpose. Two different positions of these groups were explored to allow better accommodation of the ligand in the CYP17A1 active site. Two categories of compounds were prepared: benzimidazole derivatives with the possibility of a well-established sp^2^ nitrogen–heme interaction (**3a**–**b** and **4a**–**b**), and indole derivatives which can potentially interact only with their sulfur groups (**3c**–**d** and **4c**–**d**).

The synthesis started from 4-(1*H*-benzo[*d*]imidazol-1-yl)aniline (**1a**) or 4-(1H-indol-1-yl)aniline (**1b**) which were coupled with commercially available 3-bromothioanisole **2a** and 4-bromothioanisole **2b** via a Buchwald–Hartwig reaction, obtaining sulfides **3a**–**d** (Figure 1). The reaction was carried out using third-generation (G3) Buchwald precatalysts. They are soluble in common organic solvents, and are also air, moisture, and thermally stable, making them convenient sources of the active catalyst [38]. Required anilines **1a** and **1b** were prepared according to our previously published method [39]. Briefly, indole or imidazole were first reacted with 1-fluoro-4-nitrobenze in the presence of potassium phosphate and then catalytically reduced with hydrogen over palladium. Subsequent oxidation with hydrogen peroxide afforded sulfoxide analogues **4a**–**d** [40]. This reaction was characterized by a very rapid progress and was completed in just a few minutes. Obtained sulfoxides displayed a characteristic downfield shift of the S(O)-methyl group in 1H NMR and 13C NMR (Appendix A).

### 3.2. CYP17A1 Inhibition

The initial screening of compounds showed marked inhibition of CYP17A1 hydroxylase activity. The four most potent compounds demonstrating more than 50% inhibition at a concentration of 10 µM were **3a**, **3b**, **4a**, and **4c** with remaining enzymatic activities of 23%, 31%, 35%, and 27% of the control, respectively (Figure 3A). All of them were less potent than the abiraterone used as a reference. Regarding the lyase reaction, only compound **3a** was able demonstrate an inhibitory activity higher than 50% (33% of the remaining enzymatic activity, Figure 3B). None of the compounds showed selective inhibition of CYP17A1 lyase activity. We noticed that the -meta substitution provided more active compounds compared to the ones with sulfur groups in the -para position. Contrary to this observation, we were not able to capture a clear trend when analyzing the influence of the remaining two variables: benzimidazole vs. indole moiety and sulfide vs. sulfoxide. However, it appears that benzimidazole is preferred as evidenced by **3a**, **3b**, and **4a**. Those compounds can theoretically bind to the heme via either the sp^2^ nitrogen atom or sulfur atom. This possibility does not exist for compound **4c**, which can only bind via its sulfoxide group. Still, this compound showed nearly the same level of potency as **3a**, while **4b**, which retains benzimidazole, was the least potent among the whole set. Because our tested compounds were generally less potent than the compounds they were based on, it can be concluded that the presence of the nitrogen atom seems to be superior to the sulfur atom [37].

### 3.3. Possible Binding Modes

Although the compounds bind to CYP17A1 in a similar way, adopting a conformation bending over Helix I with the benzimidazole or indole moiety facing the heme group and the sulfur part occupying a cavity formed by Helices F, G, and I (Figure 4), they also display some characteristic differences.

For benzimidazoles **3a**, **3b**, **4a**, and **4b**, the lone pair on the N3 coordinates to the iron atom in the heme group with a Fe-N3 distance between 2.2 and 2.5 Å, which is a typical distance for the binding of nitrogen-containing heteroaromatic to heme groups (Figure 4C) [32]. The indoles, **3c**, **3d**, **4c**,and **4d** do not coordinate to the heme group, but, nevertheless, the indole moiety faces the heme group with a Fe-C3 distance between 2.7 Å and 3.5 Å (Figure 4B).

As already mentioned, all the molecules bent over Helix I with the para-substituted benzene ring oriented flat against Helix I. The sulfide or sulfoxide parts of the molecules are placed in a primary hydrophobic cavity formed by Tyr201, Asn202, and Ile205 in Helix F and Arg239 in Helix G. The only specific contact between this part of the ligands and CYP17A1 is a hydrogen bond between Arg239 and the sulfoxide oxygen atom at 2.5–3.0 Å (Figure 4A).

The binding modes generated by docking suggest that the benzimidazole ring system is a better iron coordinator than the sulfoxide moiety, but it may also reflect or be influenced by the overall shape of these structurally similar ligands favoring this binding mode. We see a few examples among the poses on an inverted binding mode with the benzimidazole moiety facing Arg239, and, thereby, placing the benzene-sulfoxide part of the ligands roughly parallel to the heme group without any specific contacts to the iron atom.

### 3.4. Docking

The docking scores, which may reflect the experimental binding energies, are listed in Table 1. The benzimidazoles, **3a**, **3b**, **4a**, and **4b**, are predicted to bind better than their corresponding indoles, **3c**, **3d**, **4c**, and **4d**, with score difference of 1-4 probably due to the favorable Fe-N coordination. This calculation would corroborate the apparent better performance of the benzimidazole compounds in CYP assays. In addition to this likely significant trend in binding, we also observe some minor differences. For **3a**/**4a**, **3b**/**4b**, and **3c**/**4c**, the sulfoxides bind better than the sulfides, but for **3d**/**4d** the sulfide binds better than the sulfoxides. For the meta-substituted sulfoxides **4a** and **4c**, the *S*-enantiomer binds better than the *R*-enantiomer, whereas it is the opposite for the para-substituted sulfoxides **4b** and **4d**.

Nevertheless, we observe that, except for **4b**, the best inhibitors of the CYP17A1 hydroxylase reaction **3a**, **3b**, **4a**, and **4c** also have higher docking scores than the less efficient CYP17A1 hydroxylase inhibitors **3c**, **3d**, and **4d** (Figure 3).

### 3.5. CYP21A2, CYP3A4, and POR Inhibition

CYP21A2 is an enzyme catalyzing the transformation of progesterone and 17α-hydroxyprogesterone into mineralocorticoids and glucocorticoids, respectively (Figure 1). It is often an off-target for many CYP17A1 inhibitors due to the high structural similarity between the two enzymes [41]. To test the compounds’ selectivity, we measured CYP21A2 activity after treatment with our compounds at 10 µM. The activity was reduced to 31%, 43%, 41%, 26%, and 49% in the presence of compounds **3a**, **3b**, **4a**, **4c**, and **4d**, respectively (Figure 5A). Although the effects were not as pronounced as with abiraterone, it was nevertheless comparable to the effects observed in CYP17A1 inhibition.

CYP3A4 belongs to the CYP3A subfamily of enzymes that is responsible for the metabolism of around 30% of clinically used drugs [42]. Interactions with this enzyme leading to its inhibition or induction can result in serious drug interactions; therefore, the influence of drug candidates on these enzymes should always be investigated. Only compounds **4a** and **4c** demonstrated inhibitory activity greater than 50%, with the remaining enzymatic activity being 50% and 40%, respectively, as compared to the control (Figure 5B). Compound **3a**, which was the most potent CYP17A1 inhibitor, had little influence on CYP3A4, thereby demonstrating selectivity for CYP17A1.

Cytochrome P450 oxidoreductase (POR) plays a major role in the metabolism of drugs and steroids. All microsomal P450s depend on POR for the supply of electrons for their catalytic activities. Therefore, the disruption of POR may effectively lead to disabling CYP enzymes [43]. An interaction between POR and the compounds was investigated using resazurin, a synthetic redox substrate of POR. No significant difference in reduction of resazurin was observed in the presence of a 10 µM concentration of compounds with respect to the control (Figure 5C). Limiting the activity towards CYP3A4 and POR becomes a crucial factor in developing the compounds for targeted CYP17A1 inhibition, owing to their vast role in drug metabolism and redox reactions in the cell.

### 3.6. AKR1C3 Inhibition

AKR1C3 is an important enzyme in the androgen steroidogenic pathway [9,10]. AKR1C3 facilitates the conversion of the weak androgens—androstenedione and 5α-androstanedione—to the more active androgens, T and DHT, respectively. Its activation contributes to CRPC drug resistance in patients treated with both abiraterone and enzalutamide through increasing intracrine androgen synthesis and enhancing androgen signaling. The overexpression of *AKR1C3* in abiraterone-resistant cells suggested that its concomitant blockade might yield therapeutic response in tumors resistant to abiraterone or other CYP17A1 inhibitors [11,12]. It has been shown that the treatment of abiraterone-resistant cells with an AKR1C3 inhibitor, like indomethacin, overcomes resistance and enhances abiraterone therapy both in vitro and in vivo by reducing the levels of intracrine androgens and diminishing AR transcriptional activity [11]. The surmountable effect of combined treatment with inhibitors of AKR1C3 and abiraterone has been also observed [44,45]. Thus, the possibility of dual CYP17A1/AKR1C3 inhibition was investigated with the aim to pave the way for the future development of more potent dual inhibitors.

The indole scaffold of indomethacin is also present in ASP9521 [46], a potent AKR1C3 inhibitor that was in a phase 1 clinical trial for CRPC (May 2011, NCT01352208, discontinued) and seems to be a crucial molecular moiety for establishing effective interactions with the active site of AKR1C3 [47]. With the aim of investigating whether the indole and benzimidazole moieties present in our studied compounds could provide the ability to inhibit the AKR1C3 enzyme resulting in synergistic multi-target properties, we tested compounds **3a**–**3d** and **4a**–**4d** on the recombinant purified AKR1C3 enzyme by measuring S-tetralol oxidation in the presence of NADP^+^ (Figure 6).

Compounds **3a**–**3d** were only tested at 1 µM because they were very fluorescent at 10 µM in the enzymatic assay. Except for compound **4c**, which displayed no activity even at 10 µM, all the compounds showed weak inhibitory activity. The most active compounds were **3a** and **3c**, demonstrating that the thioether functionality is beneficial for activity, in particular when present in the meta position. If sulfide is oxidized to sulfoxide, the activity decreases, as demonstrated by **4a** versus **3a** and **4c** versus **3c**. If the thioether group is inserted in the para position, the oxidation is not detrimental to the inhibitor activity because the corresponding sulfoxides show residual inhibitory activity when tested at 10 µM. The presence of a heterocyclic ring containing a sp^2^-hybridized nitrogen atom is not essential because benzimidazoles **3a**, **3b**, **4a**, and **4b** show similar activity to their corresponding indoles **3c**, **3d**, **4c**, and **4d**, respectively.

### 3.7. Steroid Profiling

The overall effect of the compounds on hormone levels in NCI H295R cells was measured using LC-MS (Figure 7). The results correlate with enzyme assays performed using specific radiolabeled substrates. The accumulation of progesterone, comparable to abiraterone, was noted after treatment with **3a**. This might be attributed to the inhibition of CYP17A1 in the hydroxylase reaction and CYP21A2 for which progesterone is a substrate. However, compound **4c**, demonstrating a similar level of potency towards CYP17A1 and CYP21A2, was able to decrease the level of progesterone, which would be preferable in an ideal CYP17A1 inhibitor. Elevated progesterone levels have been shown to activate the proliferation of cancer cells and have oncogenic properties [48]. Notably, both compounds were less potent in both of the CYP assays compared to abiraterone. This suggest that additional mechanisms might be operational. The compounds had varying effects on the levels of glucocorticoids compared to abiraterone, but, overall, the impact was less pronounced.

This is important because one of the side effects of abiraterone treatment is glucocorticoid imbalance, necessitating the co-administration of prednisone during therapy. Interestingly, compound **3a** was able to decrease the level of DHT close to that observed after abiraterone treatment. Similarly, compounds **3b** and **4c** decreased the DHT levels two-fold compared to the control. This was also observed after abiraterone treatment, which suggests that these compounds might affect the function of 5α-reductase, which is responsible for the conversion of T to DHT. Since DHT levels were measured below the lower limit of quantification and levels of T remained relatively unchanged, it is possible that other enzymes, operating in the “backdoor” pathways where DHT is produced from androstanedione or androstanediol, are affected. The levels of weak androgens such as dehydroepiandrosterone and androstenedione were also diminished by some of the compounds, notably **3a** and **3b**. The overall effect on androgen biosynthesis, especially DHT, suggests that despite the rather weak inhibition of CYP17A1 and AKR1C3 enzymes alone, the observed result might be arising from the combined dual action.

### 3.8. Antiproliferative Activity

The AR-dependent prostate cancer cells (LNCaP) were used to evaluate the antiproliferative activity in vitro of compounds **3a**–**4b**. Although these cells are androgen sensitive, they exhibit bone metastatic behavior, better mimicking the clinical human disease [49]. After 24 h, all compounds demonstrated a weaker antiproliferative effect when compared to abiraterone (Figure 8). However, after 48 h, compound **4c** had a similar activity to that of abiraterone, decreasing the cell viability below 50%. Compound **3a** was the second most potent compound in this assay, decreasing the cell viability to close to 50%.

## 4. Conclusions

The aim of our study was twofold. The main aim was to explore the effect of CYP17A1 inhibition by compounds endowed with the sulfur moiety, and, as a secondary aim, we were interested in examining whether the dual inhibition of CYP17A1 and AKR1C3 can result in an enhanced effect on steroid hormone production. The most potent compound **3a** displayed marked inhibitory activity towards CYP17A1. **3a** also demonstrated varying degrees of inhibition against CYP21A2, CYP3A4, and POR enzymes, highlighting its potential for selective inhibition. Our limited SAR analysis suggests that the meta position of sulfur groups and the presence of benzimidazole are advantageous for the observed activity. This latter observation suggests that the presence of a nitrogen atom might be superior to a sulfur atom in inhibiting CYP17A1. Through molecular docking, we inferred that the benzimidazole ring system might serve as a better iron coordinator than the sulfur-based moiety. However, it is important to add that only limited functionalities were tested, and other motifs might offer better inhibitory profiles.

The compounds’ impact on hormone levels in NCI H295R cells was apparently stronger than what could be expected based on separate enzymatic assays alone. Our compounds did not perform particularly potently in those assays. The investigation of the potential dual inhibition of CYP17A1 and AKR1C3 offered preliminary insights, suggesting that these compounds could pave the way for the development of more potent dual inhibitors.

In essence, our findings present a promising starting point for the design of selective CYP17A1 inhibitors and potential dual inhibitors of CYP17A1 and AKR1C3. Future work will focus on refining these ideas to enhance potency and selectivity, thereby laying the ground for potential therapeutic application in hormone-related diseases, such as PCa or polycystic ovary syndrome (PCOS).

## Data Availability

Raw NMR data will be deposited in https://repod.icm.edu.pl/ (accessed on 30 August 2023).

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
