# Peer review of "Exploring the Potential of Sulfur Moieties in Compounds Inhibiting Steroidogenesis"

_biomolecules, 2023, doi:10.3390/biom13091349_

Round 1
Reviewer 1 Report
I think the work has the potential to reach interesting results but in the current status presents serious flaws. First, I do not see any clear drug design: Somehow, the premises and conclusions are disconnected. As compared to their previous work (Ref. 37) it looks like the authors want to explore the sulfur-based moieties as binders for the heme iron of the intended target in place of the nitrogen of the pyridine ring but from the docking studies seems the other way round with the in-common benzimidazole core scaffold as a better iron coordinator (I wonder why they did not carry out the docking studies prior to the design). Second, the biological outcomes are quite poor to grab the attention of the reader. I do not see interesting selectivity as well, nor the potential of developing effective dual inhibitors as claimed in the premises. Third, the number of the compounds is limited to get clear insights on these structures.
Therefore, at the moment I just can see collected experiments/studies which do not match each others.
Author Response
I think the work has the potential to reach interesting results but in the current status presents serious flaws. First, I do not see any clear drug design: Somehow, the premises and conclusions are disconnected. As compared to their previous work (Ref. 37) it looks like the authors want to explore the sulfur-based moieties as binders for the heme iron of the intended target in place of the nitrogen of the pyridine ring but from the docking studies seems the other way round with the in-common benzimidazole core scaffold as a better iron coordinator (I wonder why they did not carry out the docking studies prior to the design). Second, the biological outcomes are quite poor to grab the attention of the reader. I do not see interesting selectivity as well, nor the potential of developing effective dual inhibitors as claimed in the premises. Third, the number of the compounds is limited to get clear insights on these structures.
Therefore, at the moment I just can see collected experiments/studies which do not match each others.
We thank the reviewer for their time and valuable comments. While we agree that the potencies in separate assays are not particularly high, we would like to draw attention to the figure 7 when the observed effect of DHT level reduction is much higher than what would be expected from the isolated assays on CYPs and AKR alone (compound 3a). Even if the docking experiments provided no clear indication of the desired binding mode we wanted to test the hypothesis anyway in case the docking software does not recognize sulfur-oxygen interaction with iron, for instance. Further, with the availability of multiple different assays in our group, it is not advisable to discard new design elements based solely on computational docking studies without any functional testing. This is what we did in current manuscript to test the different design ideas and compare computational and experimental results towards designing inhibitors against CYP17A1. The notion of dual inhibition is fairly recent and we are hoping to elaborate deeper on that topic in our future publications.
Reviewer 2 Report
The manuscript can be accepted if the following concerns are addressed:
After Introduction the chemistry section says based on previous work.., a couple of line could help to bridge the gap in better explaining the rationale from the need of non-steriodal compounds to the particular kind of compounds pursued in the current work.
Did you use heptane? please confirm that it is not hexanes, why was hexanes not used which is more common and cheaply available solvent.
What was the Rf of final compounds in the given solvent systems?
for 4d why was it triturated with ether after elution with EtOAc/Heptane.
I suggest the NMR to be redrawn starting from zero ppm.
Also, I am sure the compounds which have solvent peak would have been dried, because for biological evaluation 95% pure compound is needed, can you put a cleaner NMR or show HPLC trace to confirm purity?
Author Response
The manuscript can be accepted if the following concerns are addressed:
We thank the reviewer for their time and valuable comments.
After Introduction the chemistry section says based on previous work.., a couple of line could help to bridge the gap in better explaining the rationale from the need of non-steriodal compounds to the particular kind of compounds pursued in the current work.
The beginning of the chemistry section has been rewritten as suggested by the reviewer.
Did you use heptane? please confirm that it is not hexanes, why was hexanes not used which is more common and cheaply available solvent.
We use heptane in our lab, even if it might be more expensive, mainly because it is less toxic than hexane. See the following paper: https://doi.org/10.1039/C3RA44442B and refs within.
What was the Rf of final compounds in the given solvent systems?
We did not actually measure Rfs of the final compounds. TLCs were run routinely to monitor reactions and to learn the compounds' chromatographic behavior. The choice of the solvent system was aimed to achieve Rf ~0.3. Those conditions were transferred to column chromatography for purification.
for 4d why was it triturated with ether after elution with EtOAc/Heptane.
Many times, we obtain compounds in the form of oil after chromatography. Removing traces of solvent under a high vacuum does not always result in compound solidification. In those cases, we resolve to trituration in various solvents (the compound obviously should not be soluble in a chosen solvent) in order to obtain solid form.
I suggest the NMR to be redrawn starting from zero ppm.
All NMR spectra have been redrawn as suggested. The range has been fixed to 0-10 ppm for HNMR and 0-150 ppm for CNMR and fit to highest intensity for uniform look.
Also, I am sure the compounds which have solvent peak would have been dried, because for biological evaluation 95% pure compound is needed, can you put a cleaner NMR or show HPLC trace to confirm purity?
Yes, the compounds were dried under a vacuum. The HPLC data has been added to SI. The purity was >99%.
Reviewer 3 Report
Dear authors,
A great research approach to PCa with significant results. Moreover, a well written and presented work.
Below are some remarks that you should consider:
1) In line 46 of the document there is a bold phrase stating a missing reference on spot, either you forgot to delete it or there is one reference actually missing. Action is needed.
2) In line 58 there is a “T.” before CYP11A1, is it a typo or it is stating something incompletely?
3) Lines 89-94 are referring to cases where sulfur atom is coordinating in the iron heme indeed but in your case that does not apply. Hence, I don’t see any direct advantage elaborating it so thoroughly instead of a quick reference of “additionally sulfur containing compounds have been found to coordinate…” or something else.
4) In chemistry section: All compounds are new and are characterized adequately except 3b, 4a, 4b & 4c. The later compounds are solid and missing melting points that should be provided since it’s their first entry.
5) Did the authors try to validate their models by trying to dock the TOK-001 that is co-crystalized in the PDB entry used herein? If yes, they should introduce few lines also in text and table with scoring functions.
6) Have there been any experiments performed also in regard to AKR1C3? If not or yes, say some words considering pros and cons on the dual design effort you aiming to accomplish.
7) To the future the authors should seek and test compound specificity against all important CYPs inhibitory activity (i.e. CYP1A2/2C9/2C19/2D6) including the ones tested already. Since as it is the design looks incomplete.
Best regards
Author Response
Dear authors,
A great research approach to PCa with significant results. Moreover, a well written and presented work.
We thank the reviewer for their time and valuable comments.
Below are some remarks that you should consider:
1) In line 46 of the document there is a bold phrase stating a missing reference on spot, either you forgot to delete it or there is one reference actually missing. Action is needed.
For some reason, the field was not updated by Word or it was scrambled during editing. Either way, the missing field is actually the reference to Figure 2. It has been corrected.
2) In line 58 there is a “T.” before CYP11A1, is it a typo or it is stating something incompletely?
T stands for testosterone. We have changed it to the full name as we agree this might be confusing in the figure caption.
3) Lines 89-94 are referring to cases where sulfur atom is coordinating in the iron heme indeed but in your case that does not apply. Hence, I don’t see any direct advantage elaborating it so thoroughly instead of a quick reference of “additionally sulfur containing compounds have been found to coordinate…” or something else.
We would like to keep this passage as it is. While it might seem as excessively elaborated it is actually only 5 lines of text and we feel it provides more detailed background in the introduction section.
4) In chemistry section: All compounds are new and are characterized adequately except 3b, 4a, 4b & 4c. The later compounds are solid and missing melting points that should be provided since it’s their first entry.
All final compounds were elaborated into solids even if they were initially obtained as oils after chromatography either via additional crystallization or trituration. Solids are easier to handle when performing biological assays. No melting points were taken as it is usually not required by med-chem journals. We are unable to measure it without resynthesizing additional material. We hope the NMR data together with high-res MS are enough to confirm the compounds' identity. Additionally, we included HPLC traces to demonstrate the compounds purity. All compounds were >99% pure.
5) Did the authors try to validate their models by trying to dock the TOK-001 that is co-crystalized in the PDB entry used herein? If yes, they should introduce few lines also in text and table with scoring functions.
At the beginning of this project, we confirmed the same binding pose of docked ligands as the one in the crystal structure and also performed multiple cross-docking experiments. While we extensively use different CYP17A1 PDB models in our research, we did not use TOK-001 in this study for comparison with our ligands. Our goal was to examine how sulfur-based ligands might potentially dock to the enzyme active site.
6) Have there been any experiments performed also in regard to AKR1C3? If not or yes, say some words considering pros and cons on the dual design effort you aiming to accomplish.
No docking to AKR1C3 was performed as the notion of dual inhibition is a recent one. We have more compounds in our pipeline tailored better for dual inhibition and we plan to elaborate deeper on that topic in our future publications.
7) To the future the authors should seek and test compound specificity against all important CYPs inhibitory activity (i.e. CYP1A2/2C9/2C19/2D6) including the ones tested already. Since as it is the design looks incomplete.
This is a valuable suggestion. We would like to be able to cover as many CYP isoforms as possible. However, we are limited by the available resources. We are hoping to expand the spectrum of available assays in the future with additional funding.
Best regards
Round 2
Reviewer 1 Report
Let's stay on the perspectives and potential of the work.